# Subsampled open-reference clustering creates consistent, comprehensive OTU definitions and scales to billions of sequences

Jai Ram Rideout[1,2], Yan He[3], Jose A. Navas-Molina[4], William A. Walters[5], Luke K. Ursell[6], Sean M. Gibbons[7,10], John Chase[8], Daniel McDonald[4,9], Antonio Gonzalez[9], Adam Robbins-Pianka[4,9], Jose C. Clemente[2], Jack A. Gilbert[10,11], Susan M. Huse[12], Hong-Wei Zhou[3], Rob Knight[9,13] and J. Gregory Caporaso[1,8]

[1] Center for Microbial Genetics and Genomics, Northern Arizona University, Flagstaff, AZ, USA
[2] Department of Genetics and Genomic Sciences, Icahn School of Medicine at Mount Sinai, New York, NY, USA
[3] State Key Laboratory of Organ Failure Prevention, and Department of Environmental Health, School of Public Health and Tropical Medicine, Southern Medical University, Guangzhou, Guangdong, China
[4] Department of Computer Science, University of Colorado Boulder, Boulder, CO, USA
[5] Department of Molecular, Cellular, and Developmental Biology, University of Colorado at Boulder, Boulder, CO, USA
[6] Department of Chemistry and Biochemistry, University of Colorado at Boulder, Boulder, CO, USA
[7] Graduate Program in Biophysical Sciences, University of Chicago, Chicago, IL, USA
[8] Department of Biological Sciences, Northern Arizona University, AZ, USA
[9] BioFrontiers Institute, University of Colorado at Boulder, Boulder, CO, USA
[10] Institute for Genomics and Systems Biology, Argonne National Laboratory, Lemont, IL, USA
[11] Department of Ecology and Evolution, University of Chicago, Chicago, IL, USA
[12] Department of Pathology and Laboratory Science, Warren Alpert Medical School, Brown University, Providence, RI, USA
[13] Howard Hughes Medical Institute, Boulder, CO, USA

Corresponding author
J. Gregory Caporaso,
gregcaporaso@gmail.com

## ABSTRACT

We present a performance-optimized algorithm, subsampled open-reference OTU picking, for assigning marker gene (e.g., 16S rRNA) sequences generated on next-generation sequencing platforms to operational taxonomic units (OTUs) for microbial community analysis. This algorithm provides benefits over de novo OTU picking (clustering can be performed largely in parallel, reducing runtime) and closed-reference OTU picking (all reads are clustered, not only those that match a reference database sequence with high similarity). Because more of our algorithm can be run in parallel relative to "classic" open-reference OTU picking, it makes open-reference OTU picking tractable on massive amplicon sequence data sets (though on smaller data sets, "classic" open-reference OTU clustering is often faster). We illustrate that here by applying it to the first 15,000 samples sequenced for the Earth Microbiome Project (1.3 billion V4 16S rRNA amplicons). To the best of our knowledge, this is the largest OTU picking run ever performed, and we estimate that our new algorithm runs in less than 1/5 the time than would be required of "classic" open reference OTU picking. We show that subsampled open-reference OTU picking yields results that

are highly correlated with those generated by "classic" open-reference OTU picking through comparisons on three well-studied datasets. An implementation of this algorithm is provided in the popular QIIME software package, which uses uclust for read clustering. All analyses were performed using QIIME's uclust wrappers, though we provide details (aided by the open-source code in our GitHub repository) that will allow implementation of subsampled open-reference OTU picking independently of QIIME (e.g., in a compiled programming language, where runtimes should be further reduced). Our analyses should generalize to other implementations of these OTU picking algorithms. Finally, we present a comparison of parameter settings in QIIME's OTU picking workflows and make recommendations on settings for these free parameters to optimize runtime without reducing the quality of the results. These optimized parameters can vastly decrease the runtime of uclust-based OTU picking in QIIME.

## INTRODUCTION

Three high-level strategies for defining Operational Taxonomic Unit (OTU) cluster centroids have been widely applied for centroid-based greedy clustering (*Li & Godzik, 2006*; *Edgar, 2010*) of marker gene (e.g., 16S rRNA) sequences generated on next-generation sequencing platforms to facilitate microbial community analysis. These are canonically described as *de novo*, closed-reference, and open-reference OTU picking (*Navas-Molina et al., 2013*). In each of these approaches, respectively, centroids are defined internally based only on the sequences being clustered, based only on an external, predefined database of cluster centroids, or based on a combination of the two. Each of these methods has benefits and drawbacks.

In *de novo* OTU picking, input sequences are aligned against one another, and sequences that align with greater than a user-specified percent identity are defined as belonging to the same OTU. There are many variations and free parameters in this process, such as how many alignments are performed before a sequence is assigned to an OTU or used to define a new OTU, but the common feature of these methods is that no external reference database is required. This is also the primary advantage of this method: it is not necessary to have accumulated a collection of reference sequences before working with a new marker gene. However, *de novo* OTU picking is difficult to parallelize because all processes must be able to use new OTUs that are defined by other processes. Consequently, this approach cannot scale to modern-sized data sets.

In closed-reference OTU picking, input sequences are aligned to pre-defined cluster centroids in a reference database. If the input sequence does not match any reference sequence at a user-defined percent identity threshold, that sequence is excluded. The primary advantage of closed-reference OTU picking is that it is easily parallelizable.

Because the cluster centroids are predefined, the input sequence collection can be partitioned into *n* subsets, the assignment process can be split across *n* processors, and the clustering results can be collated when all processes have completed. This dramatically reduces the "wall time" (i.e., the total time to completion as you would see it on a clock on the wall, not in terms of CPU × hours) of this method, and makes closed-reference OTU picking a convenient strategy for extremely large datasets (e.g., as in *Yatsunenko et al., 2012*). Additionally, it has the convenient feature that, because OTUs are defined by a pre-existing reference, there are typically high-quality taxonomic assignments for each OTU, and a high-quality phylogenetic tree, often based on full-length sequences rather than fragments, exists and describes the relationships among those OTUs. Furthermore, because input sequences are not compared directly to one another, but rather to an external reference, the input sequences need not overlap. This is essential, for example, if performing a meta-analysis including sequences derived from different amplification products of the same marker gene, such as the V2 and V4 regions of the 16S rRNA (e.g., as in the meta-analysis performed in *Caporaso et al., 2010*). The major drawback to closed-reference OTU picking, however, is that it cannot identify novel diversity: if a sequence has no match in the reference database, it cannot be included in the analysis, restricting analyses to already-known taxa. (Of course, the importance of this limitation decreases as the reference database increases in coverage.)

Finally, open-reference OTU picking combines the previous protocols. First, input sequences are clustered against a reference database in parallel in a closed-reference OTU picking process. However, rather than discarding sequences that fail to match the reference, these "failures" are clustered *de novo* in a serial process. Open-reference OTU picking offers benefits over both the *de novo* and closed-reference protocols. Because it includes the parallel closed-reference step, it will typically run faster than *de novo* OTU picking. And, since it includes *de novo* OTU picking of the sequences that fail to hit the reference database, all sequences are clustered, so analyses are not restricted to already-known OTUs. However, because the *de novo* clustering process is run serially, it can still be prohibitively slow for very large datasets or datasets with a substantial number of sequences that fail to hit the reference database. Because of these long runtimes, it has not yet been widely applied despite the benefits it offers.

We present a novel strategy for open-reference OTU picking that allows a larger portion of the computation to be run in parallel, which we call *subsampled open-reference OTU picking*, allowing open-reference OTU picking on very large datasets. We compare this method to "classic" open-reference OTU picking (as described in the previous paragraph) to confirm that, despite potentially slightly different OTU definitions, the summary statistics that are often used derive biological conclusions from application of these different methods to the same data set would remain the same. To achieve this, we show that alpha diversity, beta diversity, and taxonomic profiles are highly correlated between the "classic" open-reference OTU picking and subsampled open-reference OTU picking. We also compare these methods to *de novo* and closed-reference OTU picking, and explore the effect of dataset and algorithm parameters on runtime and analysis results. We note

that we specifically focus on centroid-based greedy clustering approaches in this study (e.g., as in uclust and cd-hit *Li & Godzik, 2006*; *Edgar, 2010*), not approaches that require alignment of all pairs of unique sequences (i.e., the hierarchical methods described in *Schloss & Westcott, 2011*), as the former scale better to larger data sets. However, because our full evaluation framework (metrics and data sets) and the EMP raw sequence data are all freely accessible, it is straightforward for other groups to reproduce these evaluations on alternative methods.

All analyses presented here are performed using the QIIME and pandas python packages. As far as we know, QIIME contains the only existing implementation of the subsampled open-reference OTU picking algorithm, but the algorithm is not QIIME-specific. Thus while our comparison is based on specific QIIME/uclust-based implementations of *de novo*, *closed reference*, *classic open reference*, and *subsampled open reference* OTU picking, our findings should be general to other implementations of these algorithms.

## MATERIALS AND METHODS

### Subsampled open-reference OTU picking algorithm

Open-reference OTU picking is preferable to the other methods presented here because it combines the advantages of closed-reference and *de novo* clustering. However, the *de novo* step of open-reference OTU picking can only be run serially, and therefore can be time-consuming for large datasets if many sequences fail to hit the reference database. To improve the runtime of open-reference OTU picking, we developed *subsampled open-reference OTU picking*, which incrementally increases the size of the reference database by *de novo* clustering a subset of the sequences that fail to match the reference database. The remainder of the sequences that fail to hit the reference database can then be clustered against these new cluster centroids in a parallel closed-reference OTU picking process. This allows for partial parallelization of the *de novo* clustering step and can significantly decrease runtime on large datasets, allowing open-reference OTU picking to scale to billions of input sequences (e.g., as generated in multiple Illumina HiSeq 2000 runs). It can additionally be run iteratively, so that representative sequences for the new (i.e., non-reference) OTUs can be combined with the reference database for future OTU picking runs. It is important to note that runtime is not always reduced with subsampled open-reference OTU picking. Data set and algorithm parameters have a large effect on runtime (discussed further in *Runtime differences*). This approach is similar to the Buckshot algorithm (*Cutting et al., 1992*; *Jensen et al., 2002*), initially described for semantic clustering of documents in a corpus, though we do not use the parallel hierarchical clustering approach described by *Jensen et al. (2002)* for initial clustering definition.

A detailed description of this workflow is illustrated in Fig. 1. It is implemented using uclust v1.2.22q (*Edgar, 2010*) for clustering in QIIME 1.6.0 (*Caporaso et al., 2010*) and later, though any sequence clustering software that provides support for *de novo* and closed-reference clustering could be substituted for uclust in an alternate implementation. The inputs provided to this method are demultiplexed, quality-filtered sequences, and a reference sequence collection (for example, the Greengenes 13_8 97% OTU representative

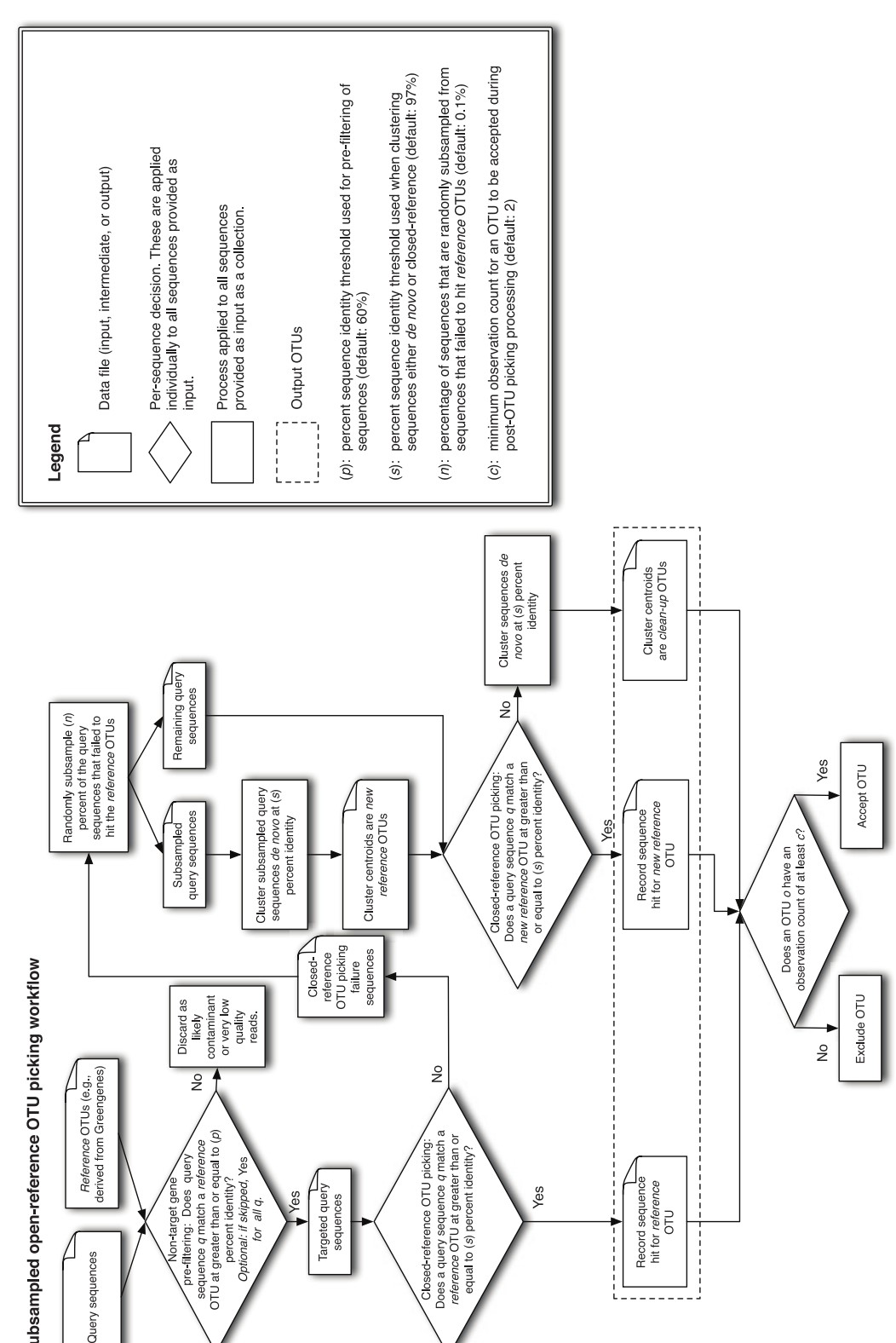

**Figure 1 Schematic of the subsampled open-reference OTU picking algorithm.**

sequences *DeSantis et al., 2006*; *McDonald et al., 2012b*). First, sequences are clustered in parallel using a closed-reference OTU picking workflow, where sequences are queried against the reference database at percent identity *s* (default 97%). If a read matches a reference sequence at greater than or equal to *s*% identity, it is assigned to the OTU defined by that reference sequence. These are referred to as the reference OTUs. Next, a random subsample of *n*% (*n* should be small, the default value in QIIME 1.8.0-dev and earlier is 0.1%) of the sequences that failed to match the reference sequence collection are clustered *de novo*, and the cluster centroids for all resulting OTUs are used to define a new reference sequence collection. Those OTUs are referred to as the new reference OTUs. The sequences that were not included in the random subsample that was clustered *de novo* then go through an additional round of parallel closed-reference OTU picking, this time where they are clustered against the new reference OTUs based on matching a sequence in the new reference sequence collection at greater than or equal to *s*% identity. This creation of a "new reference database" allows us to harness the parallelization of our closed-reference OTU picking pipeline, greatly decreasing the time it takes for sequences that fail to hit the initial reference database to be clustered into OTUs. In the final clustering step, sequences that fail to hit a reference sequence during this final closed-reference OTU picking step are clustered *de novo*. These are referred to as the clean-up OTUs. Finally, the reference OTUs, new reference OTUs, and clean-up OTUs are combined into a single OTU table (i.e., table of counts of OTUs on a per-sample basis, as described in *McDonald et al. (2012a)*), and this table, as well as a filtered table excluding OTUs with counts less than or equal to a user-defined threshold *c*, are provided to the user. By default, $c = 2$, so each OTU is observed at least twice (i.e., singleton OTUs are excluded). Because many more of the sequences can be clustered using closed-reference OTU picking in this workflow, it can run in far less time than classic open-reference OTU picking (see *Runtime differences* section below).

## Evaluation of subsampled open-reference OTU picking

We validated the subsampled open-reference OTU picking workflow by comparing it to *de novo*, closed-reference, and classic (i.e., non subsampled) open-reference clustering methods on three different datasets: the Lauber "88 Soils" study (*Lauber et al., 2009*) (referred to as *88-soils* here), the Caporaso "Moving Pictures" study (*Caporaso et al., 2011*) (referred to as *moving-pictures* here), and the Costello "Whole Body" study (*Costello et al., 2009*) (referred to as *whole-body* here) using three metrics. Table 1 provides a description of the OTU picking methods being compared. First, we tested the correlation between sample alpha diversities (OTU counts, i.e., QIIME's *observed species* metric, and Phylogenetic Diversity (PD) (*Faith, 1992*)) based on subsampled open-reference OTU picking and the other OTU picking protocols. Next, we tested whether beta diversity patterns (as determined by weighted and unweighted UniFrac (*Lozupone & Knight, 2005*) distances between samples) were consistent across OTU picking protocols, based on Mantel tests (*Mantel, 1967*) with 1,000 Monte Carlo iterations. Finally, we tested whether the same taxonomic profiles were obtained on a per-sample basis using each of the OTU picking

**Table 1 Method definitions.** Definitions of the OTU picking methods being compared here, based on the abbreviations used throughout the paper. From here, we refer to each method by its abbreviation for simplicity. We note that the both de novo (uc) and classic openreference OTU picking (ucr) are accessed through QIIME's pick_de_novo_otus.py command. ucr is applied when pick_otus:otu_picking_method uclust_ref is specified in the parameters file, and uc is applied when that option is absent. The exact command/parameter combinations used for each OTU picking run are provided in the study's GitHub repository (see Data Availability).

| Abbreviation | Title | Command | max_accepts | max_rejects | step_words | word_length | prefilter_percent_id | min_otu_size | speed_mode | Processors | reference_percent_id | subsample_fraction |
|---|---|---|---|---|---|---|---|---|---|---|---|---|
| uc | De novo | pick_de_novo_otus.py | 20 | 500 | 20 | 12 | NA | NA | slow | 1 | 0.97 | NA |
| ucr | Legacy open reference | pick_de_novo_otus.py | 20 | 500 | 20 | 12 | NA | NA | slow | 10 | 0.97 | NA |
| ucrC | Closed reference | pick_closed_reference_otus.py | 20 | 500 | 20 | 12 | NA | NA | slow | 10 | 0.97 | NA |
| ucrss | Subsampled open reference | pick_open_reference_otus.py | 20 | 500 | 20 | 12 | 0 | 1 | slow | 10 | 0.97 | 0.001 |
| ucrss_wfilter | Subsampled open reference, filtered | pick_open_reference_otus.py | 20 | 500 | 20 | 12 | 0.6 | 1 | slow | 10 | 0.97 | 0.001 |
| uc_fast | De novo, fast settings | pick_de_novo_otus.py | 1 | 8 | 8 | 8 | NA | NA | fast | 1 | 0.97 | NA |
| ucr_fast | Legacy open reference, fast settings | pick_de_novo_otus.py | 1 | 8 | 8 | 8 | NA | NA | fast | 10 | 0.97 | NA |
| ucrC_fast | Closed reference, fast settings | pick_closed_reference_otus.py | 1 | 8 | 8 | 8 | NA | NA | fast | 10 | 0.97 | NA |
| ucrss_fast | Subsampled open reference, fast settings | pick_open_reference_otus.py | 1 | 8 | 8 | 8 | 0 | 1 | fast | 10 | 0.97 | 0.001 |
| ucrss_wfilter_fast | Subsampled open reference, filtered, fast settings | pick_open_reference_otus.py | 1 | 8 | 8 | 8 | 0.6 | 1 | fast | 10 | 0.97 | 0.001 |

Table 1 (*continued*)

| Abbreviation | Title | Command | max_accepts | max_rejects | step words | word length | prefilter_percent_id | min_otu_size | speed_mode | Processors | reference_percent_id | subsample_fraction |
|---|---|---|---|---|---|---|---|---|---|---|---|---|
| ucr_fast_O29_r82 | Legacy open reference, fast settings, 82% reference OTUs, 29 processors | pick_de_novo_otus.py | 1 | 8 | 8 | 8 | 0 | 1 | fast | 29 | 0.82 | 0.001 |
| ucr_fast_O29_r97 | Legacy open reference, fast settings, 29 processors | pick_de_novo_otus.py | 1 | 8 | 8 | 8 | 0 | 1 | fast | 29 | 0.97 | 0.001 |
| ucrss_fast_O29_r82 | Subsampled open reference, fast settings, 82% reference OTUs, 29 processors | pick_open_reference_otus.py | 1 | 8 | 8 | 8 | 0 | 1 | fast | 29 | 0.82 | 0.001 |
| ucrss_fast_O29_r97 | Subsampled open reference, fast settings, 29 processors | pick_open_reference_otus.py | 1 | 8 | 8 | 8 | 0 | 1 | fast | 29 | 0.97 | 0.001 |
| ucrss_fast_O29_s1 | Subsampled open reference, fast settings, 29 processors, 1% subsample | pick_open_reference_otus.py | 1 | 8 | 8 | 8 | 0 | 1 | fast | 29 | 0.97 | 0.1 |

methods. It is important to note that we are not trying to assess whether one method is better than another using these metrics. Instead, we are testing whether the methods give highly correlated results.

### Data availability

The raw sequence data analyzed in this study is available in the QIIME Database under study numbers 103 (88-soils), 449 (whole-body), and 550 (moving-pictures). All analyses were run with QIIME 1.8.0-dev. All commands, as well as all processed data and IPython Notebooks that illustrate how to work with that data, are available in this project's GitHub repository at https://github.com/gregcaporaso/cloaked-octo-ninja.

## RESULTS AND DISCUSSION

### Subsampled versus "classic" open-reference OTU picking

Alpha diversity (Table 2; whole-body PD Pearson $r = 0.989$; 88-soils PD Pearson $r = 0.930$; moving-pictures PD Pearson $r = 0.996$), beta diversity (Table 3; whole-body unweighted UniFrac Mantel $r = 0.948$; 88-soils unweighted UniFrac Mantel $r = 0.939$; moving-pictures unweighted UniFrac Mantel $r = 0.991$) and taxonomic summaries (Table 4; whole-body: $r = 0.999$ at phylum level, 0.999 at species level; 88-soils $r = 0.999$ at phylum level, $r = 0.999$ at species level; moving-pictures $r = 0.999$ at phylum level, $r = 0.999$ at species level) were highly correlated between classic and subsampled open-reference OTU picking. Minor differences likely arise from the non-deterministic step of rarefying all samples to even sampling depth before comparing samples. These results suggest that subsampled open-reference picking yields the same results as classic open-reference OTU picking, including identical numbers of sequences failing to hit the reference database, and therefore is a suitable replacement.

### Application to the Earth Microbiome Project dataset

In order to evaluate the effectiveness of the subsampled open-reference OTU picking method on an extremely large data set, the first 15,000 samples (1.3 billion V4 16S rRNA amplicons) from the Earth Microbiome Project (EMP, *Gilbert et al., 2010*) were processed on the Amazon Web Services (AWS) EC2 platform. These samples were split across more than 60 studies, which were clustered iteratively. To the best of our knowledge, this is the largest OTU picking run ever completed. We created a StarCluster-based (http://star.mit.edu/cluster/) virtual cluster on AWS using between 8 and 18 M2.4xlarge spot instances (the number of instances was varied at different stages of the run). Each instance (or virtual cluster node) had 69 GB RAM and 8 cores. A total of 11,242 CPU hours were consumed to complete subsampled open-reference OTU picking (at 97% nucleotide identity), and the combined input and output files consumed 1.2 TB of disk space. (This runtime includes the pre-filtering step. The process would have completed much faster if this were disabled.) The resulting OTU table contained 5.6 million non-singleton OTUs. This is the largest number of OTUs identified, and the most comprehensive survey of microbial diversity across environment types to date, so it likely suggests the magnitude of the lower-bound on the microbial diversity of the Earth (although the accuracy is limited because some of

**Table 2 Alpha diversity results.** Pearson correlation coefficients (*r*) of alpha diversity for (a) 88-soils PD, (b) moving-pictures PD, (c) whole-body PD, (d) 88-soils observed species, (e) moving-pictures observed species, and (f) moving-pictures observed species.

**(a)**

|  | uc | ucr | ucrC | ucrss | ucrss_wfilter | uc_fast | ucr_fast | ucrC_fast | ucrss_fast | ucrss_fast_wfilter |
|---|---|---|---|---|---|---|---|---|---|---|
| uc | 1 | 0.951 | 0.933 | 0.934 | 0.953 | 0.956 | 0.936 | 0.927 | 0.948 | 0.947 |
| ucr | 0.951 | 1 | 0.902 | 0.931 | 0.93 | 0.946 | 0.94 | 0.903 | 0.952 | 0.944 |
| ucrC | 0.933 | 0.902 | 1 | 0.894 | 0.909 | 0.905 | 0.914 | 0.978 | 0.902 | 0.911 |
| ucrss | 0.934 | 0.931 | 0.894 | 1 | 0.929 | 0.944 | 0.935 | 0.894 | 0.948 | 0.949 |
| ucrss_wfilter | 0.953 | 0.93 | 0.909 | 0.929 | 1 | 0.952 | 0.933 | 0.903 | 0.931 | 0.943 |
| uc_fast | 0.956 | 0.946 | 0.905 | 0.944 | 0.952 | 1 | 0.953 | 0.898 | 0.956 | 0.96 |
| ucr_fast | 0.936 | 0.94 | 0.914 | 0.935 | 0.933 | 0.953 | 1 | 0.914 | 0.95 | 0.952 |
| ucrC_fast | 0.927 | 0.903 | 0.978 | 0.894 | 0.903 | 0.898 | 0.914 | 1 | 0.902 | 0.903 |
| ucrss_fast | 0.948 | 0.952 | 0.902 | 0.948 | 0.931 | 0.956 | 0.95 | 0.902 | 1 | 0.962 |
| ucrss_fast_wfilter | 0.947 | 0.944 | 0.911 | 0.949 | 0.943 | 0.96 | 0.952 | 0.903 | 0.962 | 1 |

**(b)**

|  | uc | ucr | ucrC | ucrss | ucrss_wfilter | uc_fast | ucr_fast | ucrC_fast | ucrss_fast | ucrss_fast_wfilter |
|---|---|---|---|---|---|---|---|---|---|---|
| uc | 1 | 0.996 | 0.993 | 0.996 | 0.996 | 0.995 | 0.996 | 0.992 | 0.996 | 0.996 |
| ucr | 0.996 | 1 | 0.993 | 0.997 | 0.997 | 0.995 | 0.996 | 0.992 | 0.996 | 0.997 |
| ucrC | 0.993 | 0.993 | 1 | 0.994 | 0.991 | 0.994 | 0.994 | 0.998 | 0.995 | 0.994 |
| ucrss | 0.996 | 0.997 | 0.994 | 1 | 0.996 | 0.996 | 0.997 | 0.994 | 0.997 | 0.997 |
| ucrss_wfilter | 0.996 | 0.997 | 0.991 | 0.996 | 1 | 0.994 | 0.995 | 0.991 | 0.996 | 0.996 |
| uc_fast | 0.995 | 0.995 | 0.994 | 0.996 | 0.994 | 1 | 0.997 | 0.994 | 0.997 | 0.996 |
| ucr_fast | 0.996 | 0.996 | 0.994 | 0.997 | 0.995 | 0.997 | 1 | 0.994 | 0.997 | 0.997 |
| ucrC_fast | 0.992 | 0.992 | 0.998 | 0.994 | 0.991 | 0.994 | 0.994 | 1 | 0.994 | 0.994 |
| ucrss_fast | 0.996 | 0.996 | 0.995 | 0.997 | 0.996 | 0.997 | 0.997 | 0.994 | 1 | 0.997 |
| ucrss_fast_wfilter | 0.996 | 0.997 | 0.994 | 0.997 | 0.996 | 0.996 | 0.997 | 0.994 | 0.997 | 1 |

**(c)**

|  | uc | ucr | ucrC | ucrss | ucrss_wfilter | uc_fast | ucr_fast | ucrC_fast | ucrss_fast | ucrss_fast_wfilter |
|---|---|---|---|---|---|---|---|---|---|---|
| uc | 1 | 0.985 | 0.957 | 0.985 | 0.985 | 0.984 | 0.986 | 0.961 | 0.983 | 0.984 |
| ucr | 0.985 | 1 | 0.956 | 0.99 | 0.989 | 0.988 | 0.987 | 0.96 | 0.987 | 0.986 |
| ucrC | 0.957 | 0.956 | 1 | 0.961 | 0.958 | 0.959 | 0.961 | 0.99 | 0.953 | 0.961 |
| ucrss | 0.985 | 0.99 | 0.961 | 1 | 0.991 | 0.988 | 0.99 | 0.964 | 0.989 | 0.987 |
| ucrss_wfilter | 0.985 | 0.989 | 0.958 | 0.991 | 1 | 0.985 | 0.989 | 0.963 | 0.987 | 0.985 |
| uc_fast | 0.984 | 0.988 | 0.959 | 0.988 | 0.985 | 1 | 0.986 | 0.961 | 0.986 | 0.985 |
| ucr_fast | 0.986 | 0.987 | 0.961 | 0.99 | 0.989 | 0.986 | 1 | 0.965 | 0.988 | 0.989 |
| ucrC_fast | 0.961 | 0.96 | 0.99 | 0.964 | 0.963 | 0.961 | 0.965 | 1 | 0.957 | 0.965 |
| ucrss_fast | 0.983 | 0.987 | 0.953 | 0.989 | 0.987 | 0.986 | 0.988 | 0.957 | 1 | 0.986 |
| ucrss_fast_wfilter | 0.984 | 0.986 | 0.961 | 0.987 | 0.985 | 0.985 | 0.989 | 0.965 | 0.986 | 1 |

these OTUs may be artifacts of PCR or sequencing: such artifacts, e.g., chimeras, need to be identified after the OTU picking step).

We were next interested in how long the de novo clustering step of classic open-reference OTU picking would take on the EMP data set, but as we'll illustrate this is an intractable problem in practice with current computer hardware. We began by applying de novo

Table 2 (*continued*)

**(d)**

|  | uc | ucr | ucrC | ucrss | ucrss_wfilter | uc_fast | ucr_fast | ucrC_fast | ucrss_fast | ucrss_fast_wfilter |
|---|---|---|---|---|---|---|---|---|---|---|
| uc | 1 | 0.948 | 0.88 | 0.909 | 0.924 | 0.935 | 0.934 | 0.877 | 0.925 | 0.913 |
| ucr | 0.948 | 1 | 0.905 | 0.946 | 0.947 | 0.947 | 0.953 | 0.903 | 0.938 | 0.932 |
| ucrC | 0.88 | 0.905 | 1 | 0.926 | 0.888 | 0.882 | 0.908 | 0.973 | 0.91 | 0.896 |
| ucrss | 0.909 | 0.946 | 0.926 | 1 | 0.932 | 0.923 | 0.935 | 0.915 | 0.931 | 0.929 |
| ucrss_wfilter | 0.924 | 0.947 | 0.888 | 0.932 | 1 | 0.943 | 0.946 | 0.884 | 0.932 | 0.927 |
| uc_fast | 0.935 | 0.947 | 0.882 | 0.923 | 0.943 | 1 | 0.942 | 0.883 | 0.941 | 0.94 |
| ucr_fast | 0.934 | 0.953 | 0.908 | 0.935 | 0.946 | 0.942 | 1 | 0.908 | 0.943 | 0.932 |
| ucrC_fast | 0.877 | 0.903 | 0.973 | 0.915 | 0.884 | 0.883 | 0.908 | 1 | 0.904 | 0.906 |
| ucrss_fast | 0.925 | 0.938 | 0.91 | 0.931 | 0.932 | 0.941 | 0.943 | 0.904 | 1 | 0.953 |
| ucrss_fast_wfilter | 0.913 | 0.932 | 0.896 | 0.929 | 0.927 | 0.94 | 0.932 | 0.906 | 0.953 | 1 |

**(e)**

|  | uc | ucr | ucrC | ucrss | ucrss_wfilter | uc_fast | ucr_fast | ucrC_fast | ucrss_fast | ucrss_fast_wfilter |
|---|---|---|---|---|---|---|---|---|---|---|
| uc | 1 | 0.992 | 0.984 | 0.992 | 0.992 | 0.989 | 0.99 | 0.978 | 0.989 | 0.99 |
| ucr | 0.992 | 1 | 0.994 | 0.998 | 0.998 | 0.992 | 0.997 | 0.991 | 0.997 | 0.997 |
| ucrC | 0.984 | 0.994 | 1 | 0.995 | 0.995 | 0.984 | 0.993 | 0.997 | 0.994 | 0.994 |
| ucrss | 0.992 | 0.998 | 0.995 | 1 | 0.998 | 0.992 | 0.997 | 0.991 | 0.997 | 0.997 |
| ucrss_wfilter | 0.992 | 0.998 | 0.995 | 0.998 | 1 | 0.992 | 0.997 | 0.991 | 0.997 | 0.997 |
| uc_fast | 0.989 | 0.992 | 0.984 | 0.992 | 0.992 | 1 | 0.993 | 0.981 | 0.992 | 0.992 |
| ucr_fast | 0.99 | 0.997 | 0.993 | 0.997 | 0.997 | 0.993 | 1 | 0.992 | 0.998 | 0.998 |
| ucrC_fast | 0.978 | 0.991 | 0.997 | 0.991 | 0.991 | 0.981 | 0.992 | 1 | 0.993 | 0.992 |
| ucrss_fast | 0.989 | 0.997 | 0.994 | 0.997 | 0.997 | 0.992 | 0.998 | 0.993 | 1 | 0.998 |
| ucrss_fast_wfilter | 0.99 | 0.997 | 0.994 | 0.997 | 0.997 | 0.992 | 0.998 | 0.992 | 0.998 | 1 |

**(f)**

|  | uc | ucr | ucrC | ucrss | ucrss_wfilter | uc_fast | ucr_fast | ucrC_fast | ucrss_fast | ucrss_fast_wfilter |
|---|---|---|---|---|---|---|---|---|---|---|
| uc | 1 | 0.986 | 0.971 | 0.986 | 0.986 | 0.993 | 0.988 | 0.972 | 0.988 | 0.987 |
| ucr | 0.986 | 1 | 0.984 | 0.995 | 0.995 | 0.987 | 0.993 | 0.98 | 0.993 | 0.993 |
| ucrC | 0.971 | 0.984 | 1 | 0.985 | 0.984 | 0.97 | 0.981 | 0.992 | 0.98 | 0.979 |
| ucrss | 0.986 | 0.995 | 0.985 | 1 | 0.995 | 0.987 | 0.993 | 0.981 | 0.993 | 0.992 |
| ucrss_wfilter | 0.986 | 0.995 | 0.984 | 0.995 | 1 | 0.986 | 0.993 | 0.979 | 0.992 | 0.992 |
| uc_fast | 0.993 | 0.987 | 0.97 | 0.987 | 0.986 | 1 | 0.989 | 0.972 | 0.99 | 0.988 |
| ucr_fast | 0.988 | 0.993 | 0.981 | 0.993 | 0.993 | 0.989 | 1 | 0.981 | 0.994 | 0.994 |
| ucrC_fast | 0.972 | 0.98 | 0.992 | 0.981 | 0.979 | 0.972 | 0.981 | 1 | 0.982 | 0.979 |
| ucrss_fast | 0.988 | 0.993 | 0.98 | 0.993 | 0.992 | 0.99 | 0.994 | 0.982 | 1 | 0.995 |
| ucrss_fast_wfilter | 0.987 | 0.993 | 0.979 | 0.992 | 0.992 | 0.988 | 0.994 | 0.979 | 0.995 | 1 |

clustering using the "fast" uclust parameter settings to the representative sequences from the 5.6 million non-singleton OTUs from the run described above. These representative sequences represent the full alpha diversity of the EMP data set (a property known to be important to runtime of de novo and open reference OTU clustering) but the data set contains only 5.6 m sequences, so is feasible to cluster de novo. We then subsampled this to contain between 10% and 80% of those sequences, in steps of 10% with 10 iterations at each step, and compiled the runtime for each clustering run. Figure 2

**Table 3 Beta diversity results.** Mantel correlation coefficients (*r*) of beta diversity for (a) 88-soils unweighted UniFrac, (b) moving-pictures unweighted UniFrac, (c) whole-body unweighted UniFrac, (d) 88-soils weighted UniFrac, (e) moving-pictures weighted UniFrac, and (f) moving-pictures weighted UniFrac.

**(a)**

|  | uc | ucr | ucrC | ucrss | ucrss_wfilter | uc_fast | ucr_fast | ucrC_fast | ucrss_fast | ucrss_fast_wfilter |
|---|---|---|---|---|---|---|---|---|---|---|
| uc | NA | 0.935 | 0.908 | 0.944 | 0.942 | 0.939 | 0.945 | 0.909 | 0.943 | 0.941 |
| ucr | NA | NA | 0.915 | 0.94 | 0.945 | 0.934 | 0.942 | 0.918 | 0.944 | 0.949 |
| ucrC | NA | NA | NA | 0.917 | 0.91 | 0.926 | 0.913 | 0.95 | 0.917 | 0.92 |
| ucrss | NA | NA | NA | NA | 0.94 | 0.938 | 0.945 | 0.914 | 0.938 | 0.942 |
| ucrss_wfilter | NA | NA | NA | NA | NA | 0.934 | 0.943 | 0.907 | 0.942 | 0.941 |
| uc_fast | NA | NA | NA | NA | NA | NA | 0.938 | 0.92 | 0.939 | 0.941 |
| ucr_fast | NA | NA | NA | NA | NA | NA | NA | 0.909 | 0.946 | 0.947 |
| ucrC_fast | NA | NA | NA | NA | NA | NA | NA | NA | 0.917 | 0.924 |
| ucrss_fast | NA | NA | NA | NA | NA | NA | NA | NA | NA | 0.945 |
| ucrss_fast_wfilter | NA | NA | NA | NA | NA | NA | NA | NA | NA | NA |

**(b)**

|  | uc | ucr | ucrC | ucrss | ucrss_wfilter | uc_fast | ucr_fast | ucrC_fast | ucrss_fast | ucrss_fast_wfilter |
|---|---|---|---|---|---|---|---|---|---|---|
| uc | NA | 0.992 | 0.974 | 0.988 | 0.988 | 0.992 | 0.991 | 0.977 | 0.991 | 0.992 |
| ucr | NA | NA | 0.982 | 0.992 | 0.991 | 0.991 | 0.992 | 0.984 | 0.993 | 0.993 |
| ucrC | NA | NA | NA | 0.986 | 0.985 | 0.973 | 0.982 | 0.994 | 0.981 | 0.981 |
| ucrss | NA | NA | NA | NA | 0.99 | 0.988 | 0.992 | 0.987 | 0.992 | 0.991 |
| ucrss_wfilter | NA | NA | NA | NA | NA | 0.986 | 0.99 | 0.986 | 0.99 | 0.991 |
| uc_fast | NA | NA | NA | NA | NA | NA | 0.991 | 0.976 | 0.992 | 0.991 |
| ucr_fast | NA | NA | NA | NA | NA | NA | NA | 0.983 | 0.993 | 0.992 |
| ucrC_fast | NA | NA | NA | NA | NA | NA | NA | NA | 0.982 | 0.983 |
| ucrss_fast | NA | NA | NA | NA | NA | NA | NA | NA | NA | 0.993 |
| ucrss_fast_wfilter | NA | NA | NA | NA | NA | NA | NA | NA | NA | NA |

**(c)**

|  | uc | ucr | ucrC | ucrss | ucrss_wfilter | uc_fast | ucr_fast | ucrC_fast | ucrss_fast | ucrss_fast_wfilter |
|---|---|---|---|---|---|---|---|---|---|---|
| uc | NA | 0.935 | 0.891 | 0.938 | 0.936 | 0.93 | 0.926 | 0.889 | 0.933 | 0.925 |
| ucr | NA | NA | 0.899 | 0.948 | 0.95 | 0.934 | 0.931 | 0.895 | 0.941 | 0.927 |
| ucrC | NA | NA | NA | 0.908 | 0.899 | 0.878 | 0.885 | 0.952 | 0.897 | 0.878 |
| ucrss | NA | NA | NA | NA | 0.953 | 0.938 | 0.936 | 0.905 | 0.945 | 0.928 |
| ucrss_wfilter | NA | NA | NA | NA | NA | 0.937 | 0.94 | 0.894 | 0.941 | 0.932 |
| uc_fast | NA | NA | NA | NA | NA | NA | 0.942 | 0.872 | 0.939 | 0.938 |
| ucr_fast | NA | NA | NA | NA | NA | NA | NA | 0.888 | 0.939 | 0.948 |
| ucrC_fast | NA | NA | NA | NA | NA | NA | NA | NA | 0.891 | 0.879 |
| ucrss_fast | NA | NA | NA | NA | NA | NA | NA | NA | NA | 0.933 |
| ucrss_fast_wfilter | NA | NA | NA | NA | NA | NA | NA | NA | NA | NA |

illustrates the relationship between runtime and input sequence count, along with the results of a regression analysis presenting median runtime as a function of sequence count ($r^2 = 0.98, p = 8e–6$).

In the subsampled open-reference OTU picking run on the EMP dataset, 660 million sequences failed to hit the reference database, and therefore need to be clustered de novo

Table 3 (*continued*)

**(d)**

|  | uc | ucr | ucrC | ucrss | ucrss_wfilter | uc_fast | ucr_fast | ucrC_fast | ucrss_fast | ucrss_fast_wfilter |
|---|---|---|---|---|---|---|---|---|---|---|
| uc | NA | 0.896 | 0.936 | 0.951 | 0.901 | 0.925 | 0.937 | 0.924 | 0.956 | 0.902 |
| ucr | NA | NA | 0.896 | 0.889 | 0.966 | 0.891 | 0.939 | 0.895 | 0.901 | 0.947 |
| ucrC | NA | NA | NA | 0.919 | 0.914 | 0.906 | 0.928 | 0.984 | 0.931 | 0.896 |
| ucrss | NA | NA | NA | NA | 0.9 | 0.917 | 0.947 | 0.903 | 0.949 | 0.899 |
| ucrss_wfilter | NA | NA | NA | NA | NA | 0.885 | 0.938 | 0.911 | 0.899 | 0.94 |
| uc_fast | NA | NA | NA | NA | NA | NA | 0.909 | 0.898 | 0.919 | 0.874 |
| ucr_fast | NA | NA | NA | NA | NA | NA | NA | 0.92 | 0.952 | 0.96 |
| ucrC_fast | NA | NA | NA | NA | NA | NA | NA | NA | 0.918 | 0.89 |
| ucrss_fast | NA | NA | NA | NA | NA | NA | NA | NA | NA | 0.918 |
| ucrss_fast_wfilter | NA | NA | NA | NA | NA | NA | NA | NA | NA | NA |

**(e)**

|  | uc | ucr | ucrC | ucrss | ucrss_wfilter | uc_fast | ucr_fast | ucrC_fast | ucrss_fast | ucrss_fast_wfilter |
|---|---|---|---|---|---|---|---|---|---|---|
| uc | NA | 0.971 | 0.949 | 0.97 | 0.973 | 0.972 | 0.977 | 0.949 | 0.974 | 0.966 |
| ucr | NA | NA | 0.928 | 0.952 | 0.952 | 0.957 | 0.958 | 0.928 | 0.96 | 0.954 |
| ucrC | NA | NA | NA | 0.96 | 0.94 | 0.948 | 0.934 | 0.999 | 0.965 | 0.932 |
| ucrss | NA | NA | NA | NA | 0.938 | 0.965 | 0.955 | 0.96 | 0.98 | 0.932 |
| ucrss_wfilter | NA | NA | NA | NA | NA | 0.946 | 0.966 | 0.941 | 0.951 | 0.967 |
| uc_fast | NA | NA | NA | NA | NA | NA | 0.97 | 0.948 | 0.971 | 0.949 |
| ucr_fast | NA | NA | NA | NA | NA | NA | NA | 0.934 | 0.967 | 0.967 |
| ucrC_fast | NA | NA | NA | NA | NA | NA | NA | NA | 0.965 | 0.932 |
| ucrss_fast | NA | NA | NA | NA | NA | NA | NA | NA | NA | 0.951 |
| ucrss_fast_wfilter | NA | NA | NA | NA | NA | NA | NA | NA | NA | NA |

**(f)**

|  | uc | ucr | ucrC | ucrss | ucrss_wfilter | uc_fast | ucr_fast | ucrC_fast | ucrss_fast | ucrss_fast_wfilter |
|---|---|---|---|---|---|---|---|---|---|---|
| uc | NA | 0.947 | 0.896 | 0.934 | 0.943 | 0.96 | 0.939 | 0.898 | 0.904 | 0.936 |
| ucr | NA | NA | 0.9 | 0.924 | 0.95 | 0.951 | 0.92 | 0.904 | 0.871 | 0.944 |
| ucrC | NA | NA | NA | 0.886 | 0.924 | 0.907 | 0.911 | 0.994 | 0.831 | 0.939 |
| ucrss | NA | NA | NA | NA | 0.944 | 0.92 | 0.917 | 0.882 | 0.918 | 0.911 |
| ucrss_wfilter | NA | NA | NA | NA | NA | 0.933 | 0.918 | 0.926 | 0.897 | 0.932 |
| uc_fast | NA | NA | NA | NA | NA | NA | 0.955 | 0.909 | 0.889 | 0.966 |
| ucr_fast | NA | NA | NA | NA | NA | NA | NA | 0.91 | 0.936 | 0.951 |
| ucrC_fast | NA | NA | NA | NA | NA | NA | NA | NA | 0.83 | 0.94 |
| ucrss_fast | NA | NA | NA | NA | NA | NA | NA | NA | NA | 0.866 |
| ucrss_fast_wfilter | NA | NA | NA | NA | NA | NA | NA | NA | NA | NA |

clustering in open-reference OTU picking. While it is obviously problematic to use a regression model trained on 5.6 million sequences to extrapolate the runtime on 660 million sequences, we feel that this can give us an idea of the magnitude of the runtime for the serial de novo clustering of the full dataset. Our regression model projects that the serial de novo clustering of sequences that fail to hit the reference data set would require approximately 150 days to run (in wall time). In contrast, the subsampled open-reference OTU picking run presented here (which included the pre-filtering step) ran in just under

**Table 4** **Taxonomic profile results.** Pearson correlation coefficients (*r*) of taxonomic summaries for (a) 88-soils at phylum level, (b) 88-soils at genus level, (c) moving-pictures at phylum level, (d) movingpictures at genus level, (e) whole-body at phylum level, and (f) whole-body at genus level.

**(a)**

|  | uc | ucr | ucrC | ucrss | ucrss_wfilter | uc_fast | ucr_fast | ucrC_fast | ucrss_fast | ucrss_fast_wfilter |
|---|---|---|---|---|---|---|---|---|---|---|
| uc | NA | 1 | 0.983 | 1 | 1 | 1 | 1 | 0.981 | 1 | 1 |
| ucr | NA | NA | 0.983 | 1 | 1 | 1 | 1 | 0.981 | 1 | 1 |
| ucrC | NA | NA | NA | 0.983 | 0.983 | 0.983 | 0.983 | 0.999 | 0.983 | 0.983 |
| ucrss | NA | NA | NA | NA | 1 | 1 | 1 | 0.981 | 1 | 1 |
| ucrss_wfilter | NA | NA | NA | NA | NA | 1 | 1 | 0.981 | 1 | 1 |
| uc_fast | NA | NA | NA | NA | NA | NA | 1 | 0.981 | 1 | 1 |
| ucr_fast | NA | NA | NA | NA | NA | NA | NA | 0.981 | 1 | 1 |
| ucrC_fast | NA | NA | NA | NA | NA | NA | NA | NA | 0.981 | 0.981 |
| ucrss_fast | NA | NA | NA | NA | NA | NA | NA | NA | NA | 1 |
| ucrss_fast_wfilter | NA | NA | NA | NA | NA | NA | NA | NA | NA | NA |

**(b)**

|  | uc | ucr | ucrC | ucrss | ucrss_wfilter | uc_fast | ucr_fast | ucrC_fast | ucrss_fast | ucrss_fast_wfilter |
|---|---|---|---|---|---|---|---|---|---|---|
| uc | NA | 0.939 | 0.85 | 0.939 | 0.939 | 1 | 0.94 | 0.84 | 0.94 | 0.94 |
| ucr | NA | NA | 0.821 | 1 | 1 | 0.94 | 0.998 | 0.923 | 0.998 | 0.998 |
| ucrC | NA | NA | NA | 0.821 | 0.821 | 0.85 | 0.82 | 0.818 | 0.82 | 0.82 |
| ucrss | NA | NA | NA | NA | 1 | 0.94 | 0.998 | 0.923 | 0.998 | 0.998 |
| ucrss_wfilter | NA | NA | NA | NA | NA | 0.94 | 0.998 | 0.923 | 0.998 | 0.998 |
| uc_fast | NA | NA | NA | NA | NA | NA | 0.94 | 0.84 | 0.94 | 0.94 |
| ucr_fast | NA | NA | NA | NA | NA | NA | NA | 0.921 | 1 | 1 |
| ucrC_fast | NA | NA | NA | NA | NA | NA | NA | NA | 0.921 | 0.921 |
| ucrss_fast | NA | NA | NA | NA | NA | NA | NA | NA | NA | 1 |
| ucrss_fast_wfilter | NA | NA | NA | NA | NA | NA | NA | NA | NA | NA |

**(c)**

|  | uc | ucr | ucrC | ucrss | ucrss_wfilter | uc_fast | ucr_fast | ucrC_fast | ucrss_fast | ucrss_fast_wfilter |
|---|---|---|---|---|---|---|---|---|---|---|
| uc | NA | 1 | 0.997 | 1 | 1 | 1 | 1 | 0.997 | 1 | 0.998 |
| ucr | NA | NA | 0.997 | 1 | 1 | 1 | 1 | 0.997 | 1 | 0.998 |
| ucrC | NA | NA | NA | 0.997 | 0.997 | 0.997 | 0.997 | 1 | 0.997 | 0.998 |
| ucrss | NA | NA | NA | NA | 1 | 1 | 1 | 0.997 | 1 | 0.998 |
| ucrss_wfilter | NA | NA | NA | NA | NA | 1 | 1 | 0.997 | 1 | 0.999 |
| uc_fast | NA | NA | NA | NA | NA | NA | 1 | 0.997 | 1 | 0.998 |
| ucr_fast | NA | NA | NA | NA | NA | NA | NA | 0.997 | 1 | 0.998 |
| ucrC_fast | NA | NA | NA | NA | NA | NA | NA | NA | 0.997 | 0.997 |
| ucrss_fast | NA | NA | NA | NA | NA | NA | NA | NA | NA | 0.998 |
| ucrss_fast_wfilter | NA | NA | NA | NA | NA | NA | NA | NA | NA | NA |

30 days of wall time. This illustrates that while on relatively small data sets the performance enhancement of subsampled relative to classic open-reference OTU picking is either non-existence or modest (discussed in *Run-time differences*), on datasets at the current upper limit of size, the increased parallelizability of subsampled open-reference OTU picking makes open-reference OTU picking far more tractable.

Table 4 (*continued*)

**(d)**

|  | uc | ucr | ucrC | ucrss | ucrss_wfilter | uc_fast | ucr_fast | ucrC_fast | ucrss_fast | ucrss_fast_wfilter |
|---|---|---|---|---|---|---|---|---|---|---|
| uc | NA | 0.964 | 0.929 | 0.964 | 0.963 | 0.999 | 0.923 | 0.882 | 0.923 | 0.92 |
| ucr | NA | NA | 0.963 | 1 | 0.999 | 0.967 | 0.954 | 0.923 | 0.954 | 0.951 |
| ucrC | NA | NA | NA | 0.963 | 0.963 | 0.934 | 0.925 | 0.917 | 0.925 | 0.925 |
| ucrss | NA | NA | NA | NA | 0.999 | 0.967 | 0.954 | 0.923 | 0.954 | 0.951 |
| ucrss_wfilter | NA | NA | NA | NA | NA | 0.966 | 0.953 | 0.923 | 0.953 | 0.952 |
| uc_fast | NA | NA | NA | NA | NA | NA | 0.927 | 0.887 | 0.927 | 0.924 |
| ucr_fast | NA | NA | NA | NA | NA | NA | NA | 0.885 | 1 | 0.997 |
| ucrC_fast | NA | NA | NA | NA | NA | NA | NA | NA | 0.885 | 0.884 |
| ucrss_fast | NA | NA | NA | NA | NA | NA | NA | NA | NA | 0.997 |
| ucrss_fast_wfilter | NA | NA | NA | NA | NA | NA | NA | NA | NA | NA |

**(e)**

|  | uc | ucr | ucrC | ucrss | ucrss_wfilter | uc_fast | ucr_fast | ucrC_fast | ucrss_fast | ucrss_fast_wfilter |
|---|---|---|---|---|---|---|---|---|---|---|
| uc | NA | 1 | 0.999 | 1 | 1 | 1 | 1 | 0.998 | 1 | 1 |
| ucr | NA | NA | 0.999 | 1 | 1 | 1 | 1 | 0.998 | 1 | 1 |
| ucrC | NA | NA | NA | 0.999 | 0.999 | 0.999 | 0.999 | 0.999 | 0.999 | 0.999 |
| ucrss | NA | NA | NA | NA | 1 | 1 | 1 | 0.998 | 1 | 1 |
| ucrss_wfilter | NA | NA | NA | NA | NA | 1 | 1 | 0.998 | 1 | 1 |
| uc_fast | NA | NA | NA | NA | NA | NA | 1 | 0.998 | 1 | 1 |
| ucr_fast | NA | NA | NA | NA | NA | NA | NA | 0.998 | 1 | 1 |
| ucrC_fast | NA | NA | NA | NA | NA | NA | NA | NA | 0.998 | 0.998 |
| ucrss_fast | NA | NA | NA | NA | NA | NA | NA | NA | NA | 1 |
| ucrss_fast_wfilter | NA | NA | NA | NA | NA | NA | NA | NA | NA | NA |

**(f)**

|  | uc | ucr | ucrC | ucrss | ucrss_wfilter | uc_fast | ucr_fast | ucrC_fast | ucrss_fast | ucrss_fast_wfilter |
|---|---|---|---|---|---|---|---|---|---|---|
| uc | NA | 0.959 | 0.9 | 0.959 | 0.959 | 1 | 0.913 | 0.879 | 0.913 | 0.913 |
| ucr | NA | NA | 0.918 | 1 | 1 | 0.957 | 0.967 | 0.871 | 0.967 | 0.967 |
| ucrC | NA | NA | NA | 0.918 | 0.918 | 0.896 | 0.893 | 0.935 | 0.892 | 0.893 |
| ucrss | NA | NA | NA | NA | 1 | 0.957 | 0.967 | 0.871 | 0.967 | 0.967 |
| ucrss_wfilter | NA | NA | NA | NA | NA | 0.957 | 0.967 | 0.871 | 0.967 | 0.967 |
| uc_fast | NA | NA | NA | NA | NA | NA | 0.912 | 0.876 | 0.912 | 0.912 |
| ucr_fast | NA | NA | NA | NA | NA | NA | NA | 0.855 | 1 | 1 |
| ucrC_fast | NA | NA | NA | NA | NA | NA | NA | NA | 0.854 | 0.855 |
| ucrss_fast | NA | NA | NA | NA | NA | NA | NA | NA | NA | 1 |
| ucrss_fast_wfilter | NA | NA | NA | NA | NA | NA | NA | NA | NA | NA |

## Run-time differences

The speed improvements of subsampled open-reference OTU picking arise from the fact that a larger portion of the clustering process can be parallelized. When not run in parallel, or run in parallel over only a few (e.g., 3) CPUs, classic open-reference OTU picking is likely to be faster. Similarly, for smaller data sets (e.g., less than a few million sequences), especially if most sequences have a match in the reference database (e.g., with human gut microbiome data), classic open-reference OTU picking will achieve similar runtimes to

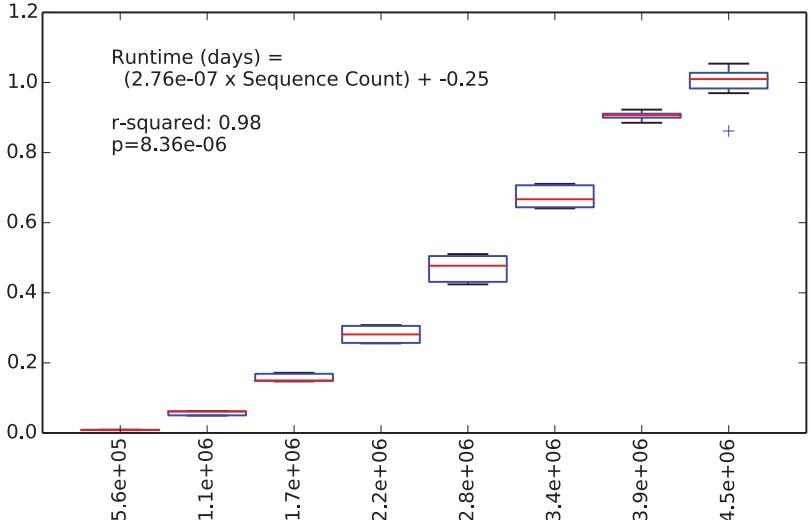

**Figure 2** **Runtime comparison.**

**Table 5** **Runtime comparison.** Comparison of runtimes (as seconds of wall time) for each method on each data set.

|  | 88-soil | Moving-picture | Whole-body |
|---|---|---|---|
| uc | 1220 | 27748 | 1095 |
| ucr | 1358 | 46576 | 1082 |
| ucrC | 226 | 28572 | 388 |
| ucrss | 1493 | 47207 | 1212 |
| ucrss_wfilter | 1885 | 76061 | 2088 |
| uc_fast | 914 | 23510 | 489 |
| ucr_fast | 1052 | 19371 | 621 |
| ucrC_fast | 44 | 2428 | 68 |
| ucrss_fast | 1021 | 23710 | 707 |
| ucrss_fast_wfilter | 1525 | 52811 | 1661 |

subsampled open-reference clustering (Table 5). However, in these cases, the results are still highly correlated, so if in doubt of which method will be faster, subsampled open-reference OTU picking is a reasonable choice as the summary statistics of interest (often alpha diversity, beta diversity and taxonomic profiles) are very unlikely to be different between the two methods.

When more sequences fail to hit the reference database, subsampled open-reference OTU picking becomes faster than classic open-reference OTU picking (Table 6). To illustrate this, we clustered the moving-pictures sequences against the 82% and 97% Greengenes reference OTUs at 97% identity using subsampled and classic open-reference OTU picking on 29 processors. When clustering against the 82% OTUs, 52.1 million failed to hit the reference, while when clustering against the 97% OTUs 3.4 million sequences failed to hit the reference. Subsampled open-reference OTU picking ran in 4000 s less wall

**Table 6 Runtime comparisons (subsampled open-reference OTU picking variants).** Comparison of runtimes (as seconds of wall time) for subsampled and "classic" open-reference OTU picking methods with variations on the default parameters.

| Abbreviation | Moving-picture |
|---|---|
| ucr_fast_O29_r82 | 21737 |
| ucr_fast_O29_r97 | 16241 |
| ucrss_fast_O29_r82 | 17812 |
| ucrss_fast_O29_r97 | 16169 |
| ucrss_fast_O29_s1 | 14911 |

time than classic open-reference clustering (in a single run of each on a system dedicated for this run time comparison) against the 82% OTUs, and in 72 s less time against the 97% OTUs, illustrating that as more sequences fail to hit the reference, subsampled open-reference OTU picking offers more of an advantage. This runtime difference would be even larger if the job were split over more processors.

Another parameter that can affect runtime of subsampled open-reference OTU picking is the size of the random subsample that is selected. The optimal setting for this parameter is affected by the size of the dataset being clustered and the diversity of the sequences that fail to match the reference database. On small datasets, or datasets with a lot of novel diversity, a large fraction (e.g., 1%) is better than a small fraction (e.g., 0.001%), but as the data set increases in size a large fraction can result in far more time spent performing *de novo* clustering of the sequences that initially fail to hit the reference database. We recommend using the default (0.1% in QIIME 1.8.0-dev and earlier), which was chosen to reduce runtime on larger datasets where optimized runtime is more important. As this parameter setting approaches zero, subsampled open-reference OTU picking becomes more like classic open-reference OTU picking, in that more of the reads that fail to hit the reference database are clustered de novo serially, and at the limit of 0% of sequences subsampled, the subsampled open reference OTU picking becomes classic open-reference OTU picking. The summary statistics investigated here are highly correlated between classic and subsampled open-reference OTU picking, suggesting that this parameter setting will not affect those statistics, but can affect runtime.

## Pre-filtering

QIIME's open-reference OTU picking workflow optionally includes a pre-filtering step, where sequences are searched against the reference database with low percent identity (the default in QIIME 1.8.0 and earlier is 60%), and sequences that fail to match are discarded from the analysis. The goal of this process is to discard sequences that are likely not representatives of the marker gene, such as host genomic sequences or products of non-specific amplification. This process is functionally similar to closed-reference OTU picking (sequence reads are searched against a pre-defined reference database), and therefore is easily run in parallel.

 

We show that alpha diversity (Table 2; whole-body PD Pearson $r = 0.991$; 88-soils PD Pearson $r = 0.930$; moving-pictures PD Pearson $r = 0.996$), beta diversity (Table 3; whole-body unweighted UniFrac Mantel $r = 0.953$; 88-soils unweighted UniFrac Mantel $r = 0.940$; moving-pictures unweighted UniFrac Mantel $r = 0.990$) and taxonomic summaries (Table 4; whole-body: $r = 1.000$ at phylum level, $r = 1.000$ at species level; 88-soils $r = 1.000$ at phylum level, $r = 1.000$ at species level; moving-pictures $r = 1.000$ at phylum level, $r = 0.999$ at species level) are highly correlated between the pre-filtered and non-pre-filtered results, when pre-filtering is performed at percent identity of 60%. Despite nearly identical results, the pre-filtering process results in vastly increased runtimes. Consequently, we no longer recommend pre-filtering of sequences prior to open-reference OTU picking. Rather, contaminant sequences should be discarded after OTU picking. This feature is now disabled by default starting with QIIME 1.8.0-dev.

One case where pre-filtering may prove useful is in the preparation of sequence data where there is a large amount of contamination of non-marker-gene sequence, for example host genomic contamination. In this case, pre-filtering can be useful to remove those sequences prior to clustering. Note that if you suspect that your sample may contain human genomic contaminant sequences, it is important to filter them out before analysis or data deposition due to Institutional Review Board or other ethical concerns related to release of human DNA sequences.

## Clustering parameters

We also investigated the effect of clustering parameters on the same summary statistics, as these can have a considerable effect on runtime. We compared uclust's default settings (referred to in QIIME as "fast mode") with the default settings in QIIME 1.8.0 and earlier ("slow mode"). We again compared the methods based on the degree to which they resulted in correlated alpha diversity (Table 2), beta diversity (Table 3), and taxonomic results (Table 4), and found that all results were highly correlated between fast and slow modes. This suggests that while fast mode will occasionally make suboptimal OTU assignments, the effects are subtle enough to be unnoticeable in downstream ecological analyses. We therefore recommend using the "fast" settings for decreased runtime, and these are now the default in QIIME 1.8.0-dev.

We do recommend using the "slow" settings if clustering sequences to build reference OTUs (for example, as is performed when building the Greengenes reference OTU collection *McDonald et al., 2012b*) because suboptimal OTU assignments can have further reaching consequences. For example, "splitting" an OTU (i.e., defining two sequences that are within $s\%$ identity of each other as the centroids of two different $s\%$ OTUs), which is always a possibility in greedy clustering algorithms, is more common with the "fast" settings than with the "slow" settings. If this occurs in a single study, the downstream effects are limited to that study and are likely only to be problematic if the split OTU is of key significance to the system being investigated. However, a split OTU when defining reference OTUs is more problematic, because those definitions will be used in many studies, increasing the chance that the split OTU will be problematic for someone. For this application,

the processing step is typically only run once per database release (which is relatively infrequent). Therefore, the longer runtime is preferable to less accurate OTU definitions in this particular application. If splitting and lumping of OTUs is of concern on your dataset, you may want to experiment with the "slow" parameter settings, which are still accessible in QIIME and we also recommend exploring the use of Oligotyping (*Eren et al., 2013*).

### Consistent OTU definitions across runs: iterative open-reference OTU picking

Subsampled open-reference clustering, as implemented in QIIME, provides new identifiers for sequences that fail to match the reference database, allowing OTUs to be directly compared across clustering runs (although sequences clustered against this expanded reference sequence collection do need to be from the same gene fragment as the sequences used to expand the reference sequence collection). These OTUs can also be used in iterative OTU picking, which is useful in studies where sequence data is continuously accumulating, for example in routine monitoring of microbial communities in human subjects (e.g., patients monitored over time), the built-environment, or during environmental clean-up.

## CONCLUSIONS

Taken together, the reduced runtime of subsampled open-reference OTU picking relative to classic open-reference OTU picking on large datasets, and the benefits that open-reference OTU picking offers over full *de novo* OTU picking (vastly decreased runtime) and closed-reference OTU picking (all sequences are clustered, not only those that match the reference collection), we recommend subsampled open-reference OTU picking when a reference collection is available.

Because the metrics provided here show that the same summary statistics are derived from the four OTU picking protocols, an interesting question is whether *de novo* or open-reference OTU picking offers any benefit over closed-reference OTU picking. The primary motivation for using methods that incorporate previously unknown OTUs (i.e., those that are not represented in the reference database) such as *de novo* and open-reference OTU picking is that OTUs not represented in the reference database might best illustrate a biological pattern of interest. For example, in the 88-soils data analyzed here, 1 of the top 10 OTUs identified as significantly different across sample pH is an OTU that is not represented in the reference database (Table 8) (this OTU was classified as in the *Actinomycetales* order by QIIME's uclust-based taxonomy classifier). Similarly, for the whole-body data set, 2 of the top 10 OTUs identified as significantly different across body sites were not represented in the reference database (these were classified as *Prevotella melaninogenica* and *Veillonella parvula* by QIIME's uclust-based taxonomy classifier). On the other hand, in the moving-pictures data analyzed here, all of the top 10 OTUs identified as significantly different across body site were OTUs represented in the reference database. Table 7 illustrates the fraction of OTUs not represented in the reference database by environment based on the Earth Microbiome Project dataset. We expect that using OTU picking methods that incorporate new OTUs is more important in samples where this fraction is higher.

**Table 7 OTU counts by environment.** Comparison of OTUs with closed-reference and open-reference OTU picking by biome in the Earth Microbiome Project dataset.

| | Average de novo OTUs (10K sequences per sample) | SD de novo OTUs (10K sequences per sample) | Average Reference OTUs (10k sequences per sample) | SD Reference OTUs (10k sequences per sample) | % novel diversity (10k seqs per sample) | % error novel diversity (10K seqs per sample) | Number of samples |
|---|---|---|---|---|---|---|---|
| Environmental Biome | | | | | | | |
| Mangrove biome | 2,169 | 1,159 | 354 | 73 | 0.86 | 0.46 | 7 |
| Tropical humid forests | 2,398 | 260 | 397 | 35 | 0.858 | 0.094 | 26 |
| Tundra biome | 1,771 | 403 | 312 | 117 | 0.85 | 0.201 | 110 |
| Deserts and xeric shrubland biome | 3,917 | 127 | 707 | 15 | 0.847 | 0.028 | 7 |
| Taiga | 2,598 | 102 | 505 | 35 | 0.837 | 0.035 | 4 |
| Marine biome | 2,040 | 1,048 | 484 | 410 | 0.808 | 0.446 | 890 |
| Aquatic biome | 714 | 299 | 177 | 199 | 0.801 | 0.403 | 762 |
| Freshwater biome | 768 | 541 | 194 | 120 | 0.798 | 0.576 | 375 |
| Warm deserts and semideserts | 2,386 | 473 | 607 | 147 | 0.797 | 0.166 | 97 |
| Tropical and subtropical moist broadleaf forest biome | 3,072 | 125 | 846 | 18 | 0.784 | 0.032 | 2 |
| Temperate needle-leaf forests or woodlands | 2,836 | 159 | 785 | 132 | 0.783 | 0.057 | 21 |
| Polar biome | 1,721 | 886 | 483 | 218 | 0.781 | 0.414 | 277 |
| Tropical and subtropical coniferous forest biome | 1,993 | 256 | 579 | 94 | 0.775 | 0.106 | 3 |
| Mixed island systems | 1,552 | 618 | 511 | 203 | 0.752 | 0.315 | 124 |
| Marginal sea | 1,795 | 325 | 611 | 225 | 0.746 | 0.164 | 7 |
| Temperate coniferous forest biome | 2,504 | 1,206 | 885 | 201 | 0.739 | 0.361 | 19 |
| Mediterranean forests, woodlands, and shrub biome | 695 | 361 | 275 | 195 | 0.717 | 0.424 | 371 |
| Large river biome | 1,844 | 629 | 743 | 369 | 0.713 | 0.282 | 5 |
| Terrestrial biome | 2,714 | 222 | 1,138 | 163 | 0.705 | 0.072 | 627 |
| Nest of bird | 821 | 276 | 355 | 138 | 0.698 | 0.262 | 313 |
| Temperate broadleaf and mixed forest biome | 1,910 | 491 | 879 | 235 | 0.685 | 0.195 | 14 |
| Temperate grasslands | 2,745 | 290 | 1,315 | 164 | 0.676 | 0.082 | 696 |
| Animal-associated habitat | 758 | 329 | 376 | 240 | 0.668 | 0.359 | 1036 |
| Mammalia-associated habitat | 973 | 357 | 583 | 222 | 0.625 | 0.27 | 1918 |
| Cold-winter (continental) deserts and semideserts | 847 | 210 | 551 | 215 | 0.606 | 0.215 | 102 |
| Temperate grasslands, savannas, and shrubland biome | 1,688 | 272 | 1,497 | 275 | 0.53 | 0.121 | 85 |
| Human-associated habitat | 292 | 242 | 590 | 366 | 0.331 | 0.498 | 1597 |

**Table 8** **Significantly different OTUs by environmental metadata.** Top 10 OTUs identified as significantly different across (a) binned pH in 88-soils, (b) body site in moving-pictures, and (c) body site in whole-body.

**(a)**

| OTU | Taxonomy | Test-statistic |
|---|---|---|
| 113212 | k__Bacteria;p__Acidobacteria;c__DA052;o__Ellin6513;f__;g__;s__ | 55.859 |
| 1123837 | k__Bacteria;p__Actinobacteria;c__Rubrobacteria;o__Rubrobacterales;f__Rubrobacteraceae; g__Rubrobacter;s__ | 50.433 |
| New.Reference OTU22 | k__Bacteria;p__Actinobacteria;c__Actinobacteria;o__Actinomycetales;f__;g__;s__ | 49.172 |
| 252012 | k__Bacteria;p__Proteobacteria;c__Gammaproteobacteria;o__Xanthomonadales;f__Sinobacteraceae;g__;s__ | 48.65 |
| 843189 | k__Bacteria;p__Acidobacteria;c__Solibacteres;o__Solibacterales;f__Solibacteraceae; g__Candidatus Solibacter;s__ | 47.006 |
| 1127423 | k__Bacteria;p__Acidobacteria;c__Acidobacteriia;o__Acidobacteriales;f__Koribacteraceae;g__;s__ | 43.87 |
| 1129210 | k__Bacteria;p__Acidobacteria;c__Acidobacteriia;o__Acidobacteriales;f__Koribacteraceae;g__;s__ | 43.804 |
| 831520 | k__Bacteria;p__Actinobacteria;c__Rubrobacteria;o__Rubrobacterales; f__Rubrobacteraceae;g__Rubrobacter;s__ | 43.625 |
| 1139779 | k__Bacteria;p__Proteobacteria;c__Alphaproteobacteria | 41.863 |
| 804187 | k__Bacteria;p__Acidobacteria;c__[Chloracidobacteria];o__RB41;f__;g__;s__ | 41.151 |

**(b)**

| OTU | Taxonomy | Test-statistic |
|---|---|---|
| 368134 | k__Bacteria;p__Firmicutes;c__Bacilli;o__Bacillales;f__Staphylococcaceae;g__Staphylococcus;s__epidermidis | 1599.696 |
| 3154070 | k__Bacteria;p__Bacteroidetes;c__Bacteroidia;o__Bacteroidales;f__Bacteroidaceae;g__Bacteroides;s__uniformis | 1625.703 |
| 1000986 | k__Bacteria;p__Actinobacteria;c__Actinobacteria;o__Actinomycetales;f__Corynebacteriaceae;g__Corynebacterium;s__ | 1630.009 |
| 1992 | k__Bacteria;p__Bacteroidetes;c__Bacteroidia;o__Bacteroidales;f__Bacteroidaceae;g__Bacteroides;s__ | 1728.164 |
| 4304475 | k__Bacteria;p__Bacteroidetes;c__Bacteroidia;o__Bacteroidales;f__Bacteroidaceae;g__Bacteroides;s__ | 1545.445 |
| 191238 | k__Bacteria;p__Firmicutes;c__Clostridia;o__Clostridiales;f__Lachnospiraceae;g__Coprococcus;s__ | 1546.436 |
| 187665 | k__Bacteria;p__Firmicutes;c__Clostridia;o__Clostridiales;f__Lachnospiraceae;g__;s__ | 1474.529 |
| 4396297 | k__Bacteria;p__Firmicutes;c__Clostridia;o__Clostridiales;f__Lachnospiraceae;g__;s__ | 1585.015 |
| 3903651 | k__Bacteria;p__Firmicutes;c__Clostridia;o__Clostridiales;f__Ruminococcaceae;g__Oscillospira;s__ | 1670.188 |
| 3472078 | k__Bacteria;p__Bacteroidetes;c__Bacteroidia;o__Bacteroidales;f__Bacteroidaceae;g__Bacteroides;s__ | 1783.488 |

In conclusion, this paper presents the performance-optimized subsampled open-reference OTU picking algorithm, now available in QIIME. This method can be applied iteratively to define stable OTUs across sequencing runs, and achieves nearly identical results to "classic" open-reference OTU picking (i.e., not including the subsampling step). It enables massive sequencing projects such as the Earth Microbiome Project to use open-reference OTU picking in far less time than is possible with classic open-reference OTU picking, which will facilitate our exploration of microbial diversity. Further, the iterative nature of the process (which is also possible with classic open-reference OTU picking) enables progressively expanding datasets, as might be generated in clinical laboratories as microbiome-based medical treatment becomes a reality, to cluster OTUs using OTU definitions from previous clustering runs as reference sequences. This

Table 8 (*continued*)

**(c)**

| | Taxonomy | Test-Statistic |
|---|---|---|
| **OTU** | | |
| 4326219 | k__Bacteria;p__Proteobacteria;c__Epsilonproteobacteria;o__Campylobacterales; f__Campylobacteraceae; g__Campylobacter;s__ | 363.881 |
| New.CleanUp. Reference OTU222 | k__Bacteria;p__Bacteroidetes;c__Bacteroidia;o__Bacteroidales;f__Prevotellaceae;g__Prevotella; s__melaninogenica | 358.02 |
| 4325533 | k__Bacteria;p__Bacteroidetes;c__Bacteroidia;o__Bacteroidales;f__Rikenellaceae;g__;s__ | 349.852 |
| New.CleanUp. Reference OTU17550 | k__Bacteria;p__Firmicutes;c__Clostridia;o__Clostridiales;f__Veillonellaceae;g__Veillonella;s__parvula | 337.656 |
| 316732 | k__Bacteria;p__Firmicutes;c__Clostridia;o__Clostridiales;f__Lachnospiraceae;g__Lachnospira;s__ | 337.309 |
| 4346374 | k__Bacteria;p__Bacteroidetes;c__Bacteroidia;o__Bacteroidales;f__Bacteroidaceae;g__Bacteroides;s__uniformis | 331.433 |
| 4458959 | k__Bacteria;p__Firmicutes;c__Clostridia;o__Clostridiales;f__Veillonellaceae;g__Veillonella | 329.772 |
| 3866487 | k__Bacteria;p__Firmicutes;c__Clostridia;o__Clostridiales;f__Lachnospiraceae;g__Oribacterium;s__ | 323.488 |
| 4391641 | k__Bacteria;p__Proteobacteria;c__Gammaproteobacteria;o__Pasteurellales; f__Pasteurellaceae;g__Haemophilus; s__parainfluenzae | 312 |
| 175751 | k__Bacteria;p__Firmicutes;c__Clostridia;o__Clostridiales;f__Lachnospiraceae;g__;s__ | 305.531 |

avoids re-clustering all sequences every time new sequences are generated, thereby vastly decreasing computational costs.

# ACKNOWLEDGEMENTS

Sample processing, sequencing and core amplicon data analysis for samples included in the Earth Microbiome Project analysis were performed by the Earth Microbiome Project (www.earthmicrobiome.org) and all amplicon and metadata has been made public through the data portal (www.microbio.me/emp).

## Funding

SMG was supported by an EPA STAR Graduate Fellowship. DM was supported in part by NSF IGERT (award number: 1144807). This work was partially supported by a grant from Arizona's Technology and Research Initiative Fund to JGC, and by a grant from the Alfred P. Sloan Foundation to JGC and RK (award number: 2012-5-42 MBRP). The funders had no role in study design, data collection and analysis, decision to publish, or preparation of the manuscript.

## Grant Disclosures

The following grant information was disclosed by the authors:
EPA STAR Graduate Fellowship.
NSF IGERT: 1144807.
Arizona's Technology and Research Initiative Fund.
Alfred P. Sloan Foundation: 2012-5-42 MBRP.

## Competing Interests

The authors declare there are no competing interests.

## Author Contributions

- Jai Ram Rideout performed the experiments, analyzed the data, contributed reagents/materials/analysis tools, prepared figures and/or tables, reviewed drafts of the paper.
- Yan He performed the experiments, analyzed the data, contributed reagents/materials/analysis tools, wrote the paper, prepared figures and/or tables, reviewed drafts of the paper.
- Jose A. Navas-Molina and Luke K. Ursell performed the experiments, analyzed the data, contributed reagents/materials/analysis tools, wrote the paper, reviewed drafts of the paper.
- William A. Walters, Sean M. Gibbons, John Chase, Daniel McDonald, Antonio Gonzalez and Adam Robbins-Pianka performed the experiments, analyzed the data, contributed reagents/materials/analysis tools, reviewed drafts of the paper.
- Jose C. Clemente conceived and designed the experiments, contributed reagents/materials/analysis tools, prepared figures and/or tables, reviewed drafts of the paper.
- Jack A. Gilbert, Susan M. Huse and Hong-Wei Zhou conceived and designed the experiments, contributed reagents/materials/analysis tools, reviewed drafts of the paper.
- Rob Knight conceived and designed the experiments, contributed reagents/materials/analysis tools, wrote the paper, reviewed drafts of the paper.
- J. Gregory Caporaso conceived and designed the experiments, performed the experiments, analyzed the data, contributed reagents/materials/analysis tools, wrote the paper, prepared figures and/or tables, reviewed drafts of the paper.

## Data Deposition

The following information was supplied regarding the deposition of related data:

https://github.com/gregcaporaso/cloaked-octo-ninja.

QIIME Database:

whole-body: ftp://thebeast.colorado.edu/pub/QIIME_DB_Public_Studies/study_449_split_library_seqs_and_mapping.zip

moving-pictures: ftp://thebeast.colorado.edu/pub/QIIME_DB_Public_Studies/study_550_split_library_seqs_and_mapping.zip

88-soils: ftp://thebeast.colorado.edu/pub/QIIME_DB_Public_Studies/study_103_split_library_seqs_and_mapping.zip.

## Supplemental Information

Supplemental information for this article can be found online at http://dx.doi.org/10.7717/peerj.545#supplemental-information.

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
