# Peer review of "Subsampled open-reference clustering creates consistent, comprehensive OTU definitions and scales to billions of sequences"

_PeerJ, doi:10.7717/peerj.545_

## Round 0.1 · original submission · Major Revisions

Both reviewers find a lot of merit in the paper, but have provided detailed reviews indicating the need for its substantial revision prior to publication.

The reviewers, in different ways, highlight some internal inconsistencies that need to be addressed, and find that the paper's presentation does not emphasise the novelty of your work clearly enough. It is also suggested, in various ways, that the paper could be shortened and focused to make it more accessible to readers.

In addition, the reviewers have suggested detailed changes. I very much hope you will agree to these changes in the paper and decide to revise it. If you do, in your covering letter with the resubmission please would you carefully explain how each of the reviewers' comments has been addressed by changes in the MS.

·

Basic reporting

No comments.

Experimental design

No comments.

Validity of the findings

Have some concern over the clarity of the central message of the paper. The paper describes a modification to an existing algorithm within the QIIME package which is used to process a particularly large dataset. The strong impression given is that performance speed has improved but the evidence presented does not back up that impression. The revised algorithm is clearly a useful addition to the QIIME package and is to be welcomed but its benefits over previous QIIME algorithms are ambiguous at best. Please see my general comments to authors for further details of my argument.

Additional comments

A modification to an existing algorithm provided by the QIIME package is presented which potentially allows for the the analysis of much larger datasets than previously. This is illustrated by applying the algorithm to 15,000 samples sequenced for the Earth Microbiome Project. Through the analysis of three smaller datasets, the authors demonstrate that results are equivalent to those generated by earlier algorithms also provided within the QIIME package but make no overt claim as to the algorithm being better.

Overall the paper is well-written and should be published as it describes a reasonable strategy for handling large datasets and give useful hints for using the authors' QIIME package. But I have a few reservations which, if addressed, would assist the reader greatly:

Firstly, let us be clear: the new "subsampled open-reference OTU picking" algorithm is an adaptation of the authors' existing "open-reference OTU picking" algorithm. The change is that those reads which do not map to the reference database are now de-novo clustered in a parallel process rather than serially as before. This is achieved by de-novo clustering a subset of the sequences that fail to map to the original reference database (a serial process) then using these to supplement the reference database allowing for a further round of (parallelized) reference matching. This, the authors contend, provides "performance-optimisation" (and hence, by implication, speed improvement) without compromising quality.
The authors demonstrate quality is not compromised by showing that results with their new algorithm are equivalent to those obtained by other strategies within the QIIME package. But they don't demonstrate that the new method is necessarily quicker. Indeed, time measurements reveal an ambiguous picture with the method sometimes preforming better, sometimes performing worse than some of the existing methods.
The authors recognise this: "It is important to note that runtime is not always reduced with subsampled open-reference OTU picking" and point out that run time is dependant on a number of parameters.

The upshot of all this is that the new method could be quicker but it just as easily might not be, depending on circumstances (e.g., more processors, character of the data). The only time that the method is clearly faster is when 1% subsample setting is used with 29 processors, and 'fast' setting (ucrss_fast_029_s1).

The authors show that their new method is able to process 15,000 records (using sufficient cpus). What they don't show is how poorly the alternative methods compare with the data so we cannot use this feat as evidence that the new algorithm is a performance improvement over the old. To be fair, the authors do not explicitly claim this - only that their new algorithm can work with a extremely large dataset. But then probably so could their other algorithms.

Where the paper is most useful is giving hints as to how to get the most out of the QIIME package as a whole. The new algorithm is a welcome addition but it doesn't necessarily perform faster than existing algorithms. The tips however, for example the value of pre-filtering, are far more performance-enhancing.

Overall, there is no evidence that the new algorithm is always faster - merely that it can do the job and might be faster under certain circumstances (contingent on various parameters). Fair enough. But I think the abstract and the text needs to make this clearer rather than leave it to the reader to make this discovery themselves having been led to believe something more substantial from choice of words. For example the conclusions careful wording implies a greater speed reduction from previously than is actually the case. The reduced runtime of the subsampled open-reference OTU picking method relative to the legacy open-reference OTU picking method is what the improvement is all about. But the "vastly decreased runtime" compared with the de novo method is the same for both the new method and the legacy method - this is not a new improvement. In a similar vein, the use of the phrase "performance-optimized" is misleading and should at least be qualified at the earliest opportunity to avoid miss-interpretation. Being more upfront in what is, and is not, claimed of the algorithm would help the reader greatly.

Final points:
I think it should be made clearer that all comparisons are made solely within the QIIME package. This is essentially a paper describing a new feature for QIIME - which is perfectly reasonable and welcome - but the authors ought to be more upfront about it.

I would ask the authors to consider whether the number of tables are justified to make one simple point: namely, that the method gives equivalent results to previous methods. I would suggest that the same point could be made by reducing the summary down to a few sentences without compromising the message (perhaps with one table to illustrate).

Lastly, I’m not sure title adequately reflects the paper and I would recommend changing it to reflect the content better. For example, Stating QIIME in the title would be very useful for those readers for whom this paper will be of genuine and welcome use.

·

Basic reporting

Start with the following definitions:
* "closed-reference" OTU picking attempts to maps each read to a sequence in a reference database and clusters reads that map to the same reference sequence
* "de novo" OTU picking runs a similarity-based clustering algorithm on the reads
* "open-reference" OTU picking first performs a closed-reference OTU picking step and runs a de novo OTU picking step on the reads that do not map to reference sequences
This paper proposes "subsampled open-reference" OTU picking, by which the authors mean the following steps:
1. run closed-reference OTU picking, let U be the unmapped reads
2. run de novo OTU picking on a random subset of U, build a reference database of the resulting cluster centers
3. run closed-reference OTU picking on U using that newly built reference database
4. run de novo OTU picking on any reads not mapped in step 3.

The idea is nice, and the paper is clearly written. The validations, given their scope, appear correctly executed. The authors have released their analysis as an iPython notebook, which is fantastic, and the method is available via the very popular QIIME analysis platform.

However, I have some reservations about the presentation of the work and the validations.

Presentation of the work:
The "subsampled open-reference" OTU picking can be factored into two distinct parts. First, there is the closed reference part of the algorithm, which is the same as classical open-reference OTU picking. Second, there is a de novo clustering step (steps 2-4 above) which first performs clustering on a random subset of the data, attempts to map the rest of the data to those clusters (which can be parallelized), and then performs de novo clustering on the rest.

It's worth doing this split because the novelty of the paper lies in the second step. This step, however, is equivalent to the so-called "Buckshot algorithm" used to initialize k-means clustering. The Buckshot algorithm was first described in a paper from Xerox PARC in 1992 [1]. The parallelization available to Buckshot algorithm in the second step is summarized in [2] with "The third phase of the Buckshot algorithm assigns the remaining documents according to their similarity to the centroids of the initial clusters. This step of the algorithm is trivially parallelized via data partitioning." I hope that the authors will describe their algorithm in terms of this previous work.

I also suggest a more neutral exploration of the idea. For example, I would suggest "classical" open-reference OTU picking rather than the pejorative "legacy", and softening phrases such as "we recommend subsampled open-reference OTU picking as the standard OTU picking protocol in all cases where a reference collection is available." I think that a change of tone is important, especially when considering the level of validation done for the method, described next.

Experimental design

Validation:
As described above, the part of the algorithm that is new to microbial ecology is using a random subset to seed clustering, which we will call "steps 2-4". Because steps 2-4 form in fact a de novo clustering algorithm, I suggest that they be evaluated as such. I would suggest following [3] in their use of normalized mutual information to compare a de novo clustering with results from a closed-reference approach.

Instead, the authors use an aggregate approach to show that results with the subsampled approach are similar to the results of other clustering techniques. The authors show, in fact, that there is a high level of correlation of *all* methods, even simply throwing away all of the reads that do not map to a reference sequence ("closed-reference" OTU picking). This in itself shows that the aggregate approach to evaluating clusters as implemented here is not sufficient to distinguish between methods of dealing with reads that do not map to the reference database. I suggest that a more appropriate way of exploring performance, if the authors really want to stay within an "aggregate" framework, is to ignore reads that map to the original reference sequences and compare "steps 2-4" as a de novo clustering approach to uclust directly. This would also equalize the data set comparisons, which are currently confounded by varying representation of sequences in existing databases (see, for example, the very high levels of correlation in the moving-pictures data set, which doubtless comes from very good representation of human gut microbiome sequences in existing databases).

In addition, the authors are not consistent with their opinion of what matters in terms of OTU clustering results. For example, they say "We do recommend using the “slow” settings if clustering sequences to build reference OTUs (for example, as is performed when building the Greengenes reference OTU collection (McDonald, Price, et al. 2012)) because suboptimal OTU assignments can have further reaching consequences." This seems in conflict with the authors' insistence elsewhere that correlation of aggregate measures is sufficient to show that an OTU picking algorithm works well.

I must be confused by my reading of line 225 that the size of the random subsample used for cluster seeding does not impact the outcome: "This parameter will not affect results, only runtime." In the limit of taking this parameter to zero, the subsampled algorithm becomes classical open-reference OTU picking. They then continue by saying "optimizing this parameter is not simple" and then incongruously give a simple general recommendation rather than exploring the results by parameter regime.

Validity of the findings

No Comments

Additional comments

Details:
28: I adore Paperpile and like that you've put in links, but you might want to note that it doesn't work at all on Firefox (and I don't think on any non-Chrome-like browsers).
36-37: the way you write this makes it sound like you are specifically talking about UCLUST's strategy. Perhaps also include other approaches that have been taken to clustering or specify your scope?
230: "However, in these cases, the results are still highly correlated, and the runtime differences are typically low enough that there is no reason to use legacy open-reference OTU picking in favor of subsampled open-reference OTU picking." Here, the authors advocate for a randomized heuristic even when the full algorithm is actually faster. (!)
278: "the same biological conclusions are derived from": no, it's the same summary statistics.

Figure 1 is difficult to follow in that the diamonds refer to a per-query-sequence question, whereas the boxes refer to actions happening on a whole collection of sequences. I'd suggest describing the questions as filters, with groups of sequences getting redirected.

Table 1 shows "De novo" and "Legacy open reference" as using the same command. Is it correct to assume that "Legacy open reference" is not currently available through QIIME?


[1] Cutting, D. R., Pedersen, J. O., Karger, D., and Tukey, J. W. Scatter/gather: A cluster-based approach to browsing large document collections. In Proceedings of 15th Annual ACM-SIGIR (1992), pp. 318–329.
[2] Jensen, Beitzel, Pilotto, Goharian, and Frieder. Parallelizing the buckshot algorithm for efficient document clustering. In CIKM '02, Proceedings of the eleventh international conference on Information and knowledge management (2002).
[3] Cai, Y. & Sun, Y. ESPRIT-Tree: hierarchical clustering analysis of millions of 16S rRNA pyrosequences in quasilinear computational time. Nucleic Acids Res. 39, e95 (2011).

---

## Round 0.2 · accepted · Accept

I would strongly recommend that you consider and address the reviewers' further few minor comments. The sugestion to re-think the paper's title seems sensible to me but I will leave the final decision to you.

·

Basic reporting

No comments

Experimental design

No comments

Validity of the findings

No comments

Additional comments

The authors have now included an estimate for the improvement in running time when their new algorithm is used with their large (15,000 sample) dataset compared with their “classic” algorithm.

This greatly enhances the manuscript: it is now clear that a decrease in running time can be substantial when very large data sets are considered. This is an important finding. This is an algorithm for handling microbiome Big Data (I'm using this possibly over-used term here in an attempt to stress the large increase in scale we're talking of here). Indeed, this surely should be the core message of the paper? Namely, with very large datasets - microbiome Big Data projects - the presented algorithm can provide real speed benefits.

For this reason I will recommend acceptance of the paper.

However I still remain doubtful over the clarity with which the authors message is being delivered. The title troubles me: it fails to communicate what seems to be of key interest to any potential reader - namely, that a new algorithm has been implemented that can greatly speed-up OTU clustering at scale without compromising on quality. As it is, the current title rather short-changes the paper, which is a shame.

I would urge the authors to reconsider the title - although I will not insist on this. I no longer think it is necessary to reference QIIME in the title - the authors argument and changes to the text remove that concern - but something along the lines of, say, “New algorithm for faster reporting of OTU definitions in microbiome Big Data projects without loss of efficiency” would, I think, signal the right message to the reader.

The results and discussion have benefited from the authors' restructuring - placing the analysis of the EMP far more prominently - this is very welcome (after all, this is the exciting bit!). This helps to focus the reader on the core message of the paper. But the tables - their repetitious nature in terms of the point being made - do not, I think, help the authors in communicating their argument. Again I will not make this a condition for my acceptance but I would ask the authors to consider whether this information - which I fully accept will be useful to some readers - could be presented less prominently, e.g., as supplementary data, so as not to diminish their message for the majority of readers.

In summary, the changes the authors have made are very welcome and I am happy to recommend acceptance. However, I feel the clarity of the paper would benefit in the ways outlined above.

·

Basic reporting

The exposition of this paper is much improved in this revision.

I suggest that it be made more clear in the abstract that the intent is to enable community diversity analysis using this new method. For example, in the first sentence I suggest replacing "microbial community analysis" with "microbial diversity analysis."

Minor comment:

263: putting something between the two instances of "subsampled" would clarify things.

Experimental design

This paper meets the experimental design guidelines.

Validity of the findings

The authors have clearly shown that a certain type of diversity analysis is enabled for very large data sets by this method.

Additional comments

Figure 1 is improved with the addition of an explanatory legend. I still think it a little strange that the most common outcome will be to have a data set go through both the "yes" and "no" directions for different sequences, but I think the intent is comprehensible.